# Damage Mechanism of Cu_6_Sn_5_ Intermetallics Due to Cyclic Polymorphic Transitions

**DOI:** 10.3390/ma12244127

**Published:** 2019-12-10

**Authors:** Zhihao Zhang, Cunwei Wei, Huijun Cao, Ye Zhang

**Affiliations:** 1Shenzhen Research Institute of Xiamen University, Shenzhen 518055, Chinazhangye@xmu.edu.cn (Y.Z.); 2Fujian Key Laboratory of Advanced Materials, College of Materials, Xiamen University, Xiamen 361005, China; 3School of Mechanical and Automation Engineering, Xiamen City University, Xiamen 361005, China

**Keywords:** Cu_6_Sn_5_ intermetallic, solid state polymorphic transition, superstructure, transmission electron microscopy, damage mechanism, multiple reflows

## Abstract

The formation of high-melting-point Cu_6_Sn_5_ interconnections is crucial to overcome the collapse of Sn-based micro-bumps and to produce reliable intermetallic interconnections in three-dimensional (3D) packages. However, because of multiple reflows in 3D package manufacturing, Cu_6_Sn_5_ interconnections will experience cyclic polymorphic transitions in the solid state. The repeated and abrupt changes in the Cu_6_Sn_5_ lattice due to the cyclic polymorphic transitions can cause extreme strain oscillations, producing damage at the surface and in the interior of the Cu_6_Sn_5_ matrix. Moreover, because of the polymorphic transition-induced grain splitting and superstructure phase formation, the reliability of Cu_6_Sn_5_ interconnections will thus face great challenges in 3D packages. In addition, the Cu_6_Sn_5_ polymorphic transition is structure-dependent, and the *η*′↔*η* polymorphic transition will occur at the surface while the *η*′↔*η_s_*↔*η* polymorphic transition will occur in the deep matrix. This study can provide in-depth understanding of the structural evolution and damage mechanism of Cu_6_Sn_5_ interconnections in real 3D package manufacturing.

## 1. Introduction

Three-dimensional (3D) packages, which can merge chip technology and packaging technology together, have great application in overcoming the scaling limits in 2D integrated circuits (ICs). The most typical characteristics of this technology are to vertically stack the multi-chips in a tight device space and to continuously minimize the interconnections on a limited chip area. As shown in Figure 1, there are four kinds of solder joints in 3D packages. The ball grid array (BGA) solder joints of 760 µm in diameter are above the printed circuit board (PCB). The controlled collapse chip connection (C4) solder joints of 250 µm in diameter are above the BGA package substrate. The Sn-based micro-bumps (Sn-bumps) of 80 µm in diameter are above the interposer substrate. The Sn-bumps of 20 µm in diameter are located between the stacked chips and through-Si-via (TSV) structures. In 3D packages, the Sn-bumps of 20 µm in diameter must be completely transformed into the intermetallic (IMC) phase in the joints after the first reflow, to avoid joint collapse during multiple reflow processes [1]. Accordingly, the IMCs, e.g., Cu_6_Sn_5_, Cu_3_Sn and Ni_3_Sn_4_, are quickly emerging as innovative high-temperature interconnection materials due to their unique abilities to join at low temperatures and operate at high temperatures [2,3,4,5]. Among these IMCs, Cu_6_Sn_5_ has received a substantial amount of attention because of its fast growth rate and high-temperature strength [6,7]. Sakuma et al. even concluded that realization of the Cu_6_Sn_5_ interconnections may be a guarantee of manufacturability and reliability in 3D packages [8].

Because of the thermal mismatch between Cu_6_Sn_5_ and Si during multiple reflow processes, cyclic strains are considered to be the dominant cause of damage to IMC interconnections in 3D packages [9,10]. However, Cu_6_Sn_5_ has at least two phases in the solid state, and its *η* phase (*P6_3_/mmc*) can thermodynamically transform to the *η*′ phase (*C2/c*) with a volume expansion of 2.2% below 186 °C [11,12]. Because small joints have elevated heating/cooling rates, the polymorphic transition of *η*′↔*η* may be kinetically retarded owing to the insufficient transition time. It is concerning that this retarded polymorphic transition may cause a transition stress that continuously damages the Cu_6_Sn_5_ IMC interconnection [13,14]. Although some researchers have begun to explore the effect of the Cu_6_Sn_5_ polymorphic transition on the interconnect reliability [14,15,16], the intrinsic damage mechanism has seldom been reported during multiple reflow processes. On 7 October, 2019, Samsung Electronics announced that it had developed the industry’s first 12-layer-3D-IC, and that the IMC interconnections in such 3D IC may experience over 10 reflow cycles. Because the number of stacking layers will never end, a 50-layer 3D IC may be fabricated in the near future, and the IMC interconnections in such 3D IC may experience 50 or even more reflow cycles. Therefore, the determination of the damage mode of Cu_6_Sn_5_ during multiple reflow processes is urgent in real 3D package manufacturing. The aims of this study were to explore the structural evolution of the Cu_6_Sn_5_ under multiple reflow processes, to study the damage mechanism induced by the cyclic polymorphic transitions, and to confirm the influence of cyclic polymorphic transitions on the reliability of Cu_6_Sn_5_ interconnections in 3D package manufacturing.

## 2. Materials and Methods 

The Cu_6_Sn_5_ IMCs were separated via a precipitation method from a supersaturated Sn–Cu melt [17]. To be specific, a solder bath containing 1 kg of the Sn-0.7 wt% Cu solder and 30 g of the fine Cu powder was placed in a furnace, heated to 350 °C for 24 h under N_2_ atmosphere. After the Cu powder was completely dissolved, the solder was quickly cooled down to 250 °C and maintained at this temperature for 10 h. Several rod-type IMCs were then separated from this solder using a quartz screen and quenched down to 0 °C. To expose the pure IMCs, the residual solder on their surfaces was etched by a 10% hydrochloric acid alcohol solution.

Subsequently, the as-prepared IMCs were reflowed under a ramp-soak-spike profile provided by a BGA rework machine (RD-500II, DEN-ON INSTRUMENTS Inc., Tokyo, Japan). The temperature ranges of the ramp zone, soak zone and spike zone were set at 20–170 °C, 170–200 °C, and 200–220 °C, respectively, with the same ramp rates of 10 °C/min. After heating to 220 °C, the samples were maintained at this temperature for 10 min, and then cooled down to 20 °C with a cooling rate of 20 °C/min. The aforesaid reflow process was repeated 1 time, 10 times, and 50 times. 

The structural evaluation of the Cu_6_Sn_5_ IMCs after multiple reflows was observed by scanning electron microscopy (SEM, SU-70, Hitachi Inc., Fukuoka, Japan). Moreover, the Cu_6_Sn_5_ grains that were reflowed 1 time were milled to obtain a fine powder and studied by X-ray diffraction (XRD, D8 Advance, Bruker Inc., Karlsruhe, Germany, Cu-*Kα* radiation) and differential scanning calorimetry (DSC, 214 Polyma, NETZSCH Inc., Selb, Germany). It should be noted that the XRD measurement was conducted at a cooling rate of 20 °C/min from 220 °C to 20 °C, and that the data at each selected temperature were collected in the 2*θ* range of 30°–55° with a scan rate of 1°/min; the DSC measurement was conducted at a heating/cooling rate of 1 °C/min from 20 °C to 220 °C under an air atmosphere, and the sample of ~28 mg was involved. In addition, to study the *η*′↔*η* polymorphic transition, two Cu_6_Sn_5_ grains that were reflowed 1 time and 50 times, respectively, were cut by a focused-ion beam (FIB, NanoLab 600i, FEI Inc., Hillsboro, OR, USA), and the corresponding microstructures were studied by transmission electron microscopy (TEM, Tecnai G2 F20, Philips Inc., Eindhoven, Holland). It should be noted that Figure 2 shows a flowchart of the preparation, reflow tests, and the characterization of Cu_6_Sn_5_ IMCs.

## 3. Results and Discussion

Figure 3 shows the surface evolution of the Cu_6_Sn_5_ IMCs with increasing reflow time. After being reflowed 1 time, the Cu_6_Sn_5_ grains maintained a rod-like shape with flat and smooth surfaces. The diameters of these grains were in the range of 20–25 µm, and each grain was a single crystal based on the previous study [17]. Interestingly, after being reflowed 10 times, although the overall structures of these IMC grains did not change, the original smooth surfaces became rough and full of microcracks (Figure 3b). After being reflowed 50 times, the Cu_6_Sn_5_ surfaces became damaged and fragmented, and numerous protrusion-like structures with irregular shapes and sizes emerged in certain depressed areas (Figure 3c). Apparently, the surface damage of the Cu_6_Sn_5_ grains arose after multiple reflows, and the damage extent seemed to increase with the increasing number of reflow times. Hence, we reasonably hypothesize that multiple reflows will produce an adverse effect on the reliability of Cu_6_Sn_5_ interconnections in 3D packages. To verify our speculation, the structural changes inside the Cu_6_Sn_5_ grains should be further explored.

Figure 4a shows the XRD patterns of the Cu_6_Sn_5_ powder after being reflowed 1 time at selected temperatures. The patterns in the temperature range of 20–150 °C are roughly identical, and by comparing the peak positions and peak intensities with the reference database, it is concluded that the corresponding main diffraction peaks are attributable to the *η*′ phase (International Centre for Diffraction Data (ICDD) 45-1488, space group of *C2*/*c*). The XRD patterns in the temperature range of 170–220 °C are similar, and the corresponding main diffraction peaks are attributable to the *η* phase (ICDD 47-1575, space group of *P6_3_/mmc*). Moreover, there are numerous relatively weak peaks at 2*θ* values of 32.35°, 39.48°, 48.39° and 49.84° for the patterns in the temperature range of 20–150 °C (marked by the dark-blue squares) and at 2*θ* values of 38.50°, 46.12° and 47.43° for the patterns in the temperature range of 170–220 °C (marked by the red circles), which cannot be well matched to the standard XRD files of the Cu_6_Sn_5_. Because superstructures may be created in the Cu_6_Sn_5_ during the polymorphic transition of *η*′↔*η* [12], these peaks are very likely structural in origin. In addition, in the 2*θ* range of 42°–44°, there are two main diffraction peaks (at ~42.82° and ~43.32°) for the patterns in the temperature range of 170–220 °C, but one (at ~42.82°) gradually disappears at decreasing temperatures of 20–150 °C. The main diffraction peaks have fundamentally changed, implying that a reconfiguration of the Cu_6_Sn_5_ orientations may occur after the *η*′↔*η* polymorphic transition.

Figure 4b shows the unit cell volumes of the Cu_6_Sn_5_ (*V*) at selected temperatures calculated from our XRD data. Two almost linear increases in *V* are detected, with increasing temperatures ranging from 20 to 150 °C and 170 to 220 °C, and a sharp fall of 0.72% occurs from 150 to 170 °C. The former is in accordance with thermal expansions and contractions, and the latter should originate from the Cu_6_Sn_5_ polymorphic transition. Because the theoretical value of *V* is 781.71 Å^3^, the difference between the theoretical and actual values in *V* will inevitably produce the tensile/compressive strains inside the Cu_6_Sn_5_ lattice. Especially during multiple reflows, a repeated abrupt change in *V* due to the cyclic polymorphic transitions may cause strain oscillations at the micro level. Moreover, at the macro or meso level, the transition strain may be partially released from the grain surface and be partially accumulated inside the grain matrix. Finally, the oscillation, release and accumulation of the transition strain will produce the structural damage of the Cu_6_Sn_5_, leading the Cu_6_Sn_5_ grains to crack (Figure 3b) or even break (Figure 3c).

Figure 5 shows the DSC heat flow curves of the Cu_6_Sn_5_ powder after being reflowed 1 time. Two endothermic peaks (at ~181.1 °C and ~187.1 °C, respectively) are detected during heating, while two exothermic peaks (at ~148.4 °C and ~162.3 °C, respectively) are detected during cooling. The distribution of the two latent heat peaks in the heating/cooling curve reflects that there should be three phases participating in the process. In other words, besides the *η*′ and *η* phases, a new phase (defined as the *η_s_* phase) may be involved in the Cu_6_Sn_5_ polymorphic transition, which is consistent with our XRD forecast. Undoubtably, if the *η_s_* phase indeed exists, the first and second latent heat peaks will be related to the *η*′↔*η_s_* and *η_s_*↔*η* phase transitions, respectively. Therefore, a single-phase state of the Cu_6_Sn_5_ powder during cooling can be predicted: e.g., the *η* phase at above 170 °C, the *η_s_* phase at close to 150 °C, and the *η*′ phase at below 140 °C. However, based on our XRD data, the *η_s_* phase seems always to be associated in the *η*′ or *η* phase, rather than existing as a separate phase. Hence, an exploration of the existence of the *η_s_* phase is urgently required. 

Figure 6a shows the cross-section microstructure of a Cu_6_Sn_5_ grain after being reflowed 1 time. No grain boundary exists in the Cu_6_Sn_5_ matrix, proving that the original rod-type grain before FIB cutting is indeed single-crystal. As shown in Figure 6b, the selected area diffraction pattern (SADP) analysis confirmed that the entire Cu_6_Sn_5_ grain is the *η*′ phase; the normal direction of the matrix is along the [$\bar{2}$01]*_η_*_′_ direction, and the external surface belongs to the ($\bar{3}$3$\bar{6}$)*_η_*_′_ plane. Interestingly, after being reflowed 50 times (Figure 7a), the original Cu_6_Sn_5_ single-crystal matrix has split into small grains whose grain boundaries are extended from the external surface to the deep matrix. The grains located close to the external surface (e.g., Grain A and Grain B) seem larger than those located deep inside the matrix (e.g., Grain C and Grain D). Based on the corresponding SADPs in Figure 7b, both Grain A and Grain B are identified as the *η*′ phase, and a set of orientation relationships are determined as follows: [010]*_η_*_′-A_//[$\bar{2}$01]*_η_*_′-B_, ($\bar{4}$02)*_η_*_′-A_//(336)*_η_*_′-B_ and (204)*_η_*_′-A_//(1$\bar{3}$2)*_η_*_′-B_. The orientation of Grain B is highly similar to the orientation of the Cu_6_Sn_5_ matrix in Figure 6b, but the orientation of Grain A has changed significantly. On the one hand, this indicates that the newly generated grains should originate from the original Cu_6_Sn_5_ single-crystal matrix; on the other hand, it demonstrates that the reconfiguration of the Cu_6_Sn_5_ orientations has indeed occurred after multiple reflows, at least for certain grains.

Notably, the theoretical interplanar spacings of the (204)*_η_*_′_, (1$\bar{3}$2)*_η_*_′_, (010)*_η_*_′_ and ($\bar{2}$01)*_η_*_′_ planes are 2.103 Å, 2.103 Å, 7.294 Å and 5.103 Å, respectively; thus, the lattice mismatch between every one (204)*_η_*_′_ plane and every one (1$\bar{3}$2)*_η_*_′_ plane is 0%, while the potential lattice mismatch between every two (010)*_η_*_′_ planes and every three ($\bar{2}$01)*_η_*_′_ planes is −4.71%. Despite such small lattice mismatches, some nanovoids are indeed generated at the boundary between Grain A and Grain B after being reflowed 50 times (Figure 7c). Therefore, multiple reflows can indeed result in the damage to inner Cu_6_Sn_5_ grains and should be harmful to the reliability of the Cu_6_Sn_5_ interconnections in the 3D package. More importantly, as shown in Figure 7b, Grain C has a five-fold superlattice of Grain A running along the [$\bar{4}$0$\bar{2}$]*_η_*_′_ direction, while Grain D has a five-fold superlattice of Grain B running along [060]*_η_*_′_. This is direct evidence that a superstructure phase of *η_s_* indeed exists in the Cu_6_Sn_5_ matrix, and this finding can perfectly support our speculations in both the XRD and DSC studies.

The formation of the superstructure phase is always strain-induced and structure-dependent (e.g., a strain-induced superstructure phase was observed in ZrCu during the martensitic transition [18]). Following this rule, the formation of the superstructure *η_s_* phase and the mechanism of Cu_6_Sn_5_ polymorphic transition can be inferred as below (Figure 8a). When the temperature satisfies the requirement of Cu_6_Sn_5_ phase transition, the *η*′↔*η* polymorphic transition will begin. Due to the rapid volume change during this transition, the transition strain will inevitably be generated in the Cu_6_Sn_5_ lattice; subsequently, such strain will be partially released from the external surface to form a strain-free layer and also be partially accumulated inside the matrix to create a strain-accumulation zone. From the point of view of phase transition energy, the nearer to the surface, the smaller strain energy (i.e., the main resistance to phase transition in solid state) will be, and phase transition is more prone to occur. Hence, the normal *η*′↔*η* polymorphic transition can be detected at or close to the external surfaces of the Cu_6_Sn_5_ grains, just as observed in Somidin’s in situ heating TEM experiments [11,12]. However, when the strain energy between *η*′ and *η* phases is larger than the volume free energy change between *η*′ and *η* phases (i.e., the driving force of phase transition), the *η*′↔*η* polymorphic transition may stop, and the accumulated strains will promote the metastable phase formation in order to decrease the threshold of *η*′↔*η* phase transition. Hence, the *η*′↔*η_s_*↔*η* polymorphic transition will occur inside the Cu_6_Sn_5_ deep matrix, just as we detected in Figure 7b. Because the *η_s_* phase is metastable and only exists inside the matrix, it can thus be detected in and accompanied by the *η*′ or *η* phase, just as shown in Figure 4a.

In addition, because the *η*→*η*′ phase transition is an expansion transition, we can infer that the *η*→*η_s_*→*η*′ phase transition is also an expansion transition. Based on Figure 4b, the compressive strain will exist in the Cu_6_Sn_5_ matrix during cooling below 200 °C; thus, the *η_s_* phase observed in Figure 7b seems to be created only by the compressive strain, and the occurrence of this phase seemed to be in the range of 150–170 °C based on Figure 5. Accordingly, as shown in Figure 8b, the compressive strain generated by the *η*→*η*′ expansion transition will create the strain energy inside the matrix, producing the delayed occurrence of the *η*→*η_s_* expansion transition (i.e., the exothermic peak of *η*→*η_s_* will shift to the left); similarly, the compressive strain generated by the *η*→*η_s_* expansion transition may increase the strain energy inside the deep matrix, and the exothermic peak of *η_s_*→*η*′ will shift further to the left. Therefore, there should be three exothermic peaks involved in the Cu_6_Sn_5_ phase transition during the cooling process. Notably, the compressive strain induced by the thermal expansion of the Cu_6_Sn_5_ during the heating process will result in a premature occurrence of the *η*′→*η_s_*→*η* shrinkage transition; thus, there should also be three endothermic peaks for the Cu_6_Sn_5_ phase transition during the heating process. Finally, when the strain is large enough, the splitting of the Cu_6_Sn_5_ grains will occur. Accordingly, the strain induced by the cyclic Cu_6_Sn_5_ phase transitions is the dominant cause of the crack generation that damages the Cu_6_Sn_5_ grains under multiple reflow processes. A further study may verify whether the *η_s_* phase is just one of the *η*^6^ phase, *η*^8^ phase and *η*^4+1^ phase proposed in Nogita’s study [19,20], and evaluate its lattice parameters. 

## 4. Conclusions

The surface and inner structural damage to Cu_6_Sn_5_ IMCs after multiple reflow processes was explored in this study. The results show that the superstructure phase formation occurs in Cu_6_Sn_5_ due to the accumulation of the strains induced by cyclic phase transitions. Moreover, the nanovoids can be generated at the grain boundaries owing to the lattice mismatch, producing the splitting of the Cu_6_Sn_5_ grains after multiple reflows. In addition, the *η*′↔*η* polymorphic transition will occur at the Cu_6_Sn_5_ surface while the *η*′↔*η_s_*↔*η* polymorphic transition will occur in the deep matrix. It can be predicted that, if the Cu_6_Sn_5_ polymorphic transition cannot be limited effectively during multiple reflows, it may become one of the most dominant causes of failure for Cu_6_Sn_5_ IMC joints in 3D packages.

## Figures and Tables

**Figure 1 materials-12-04127-f001:**
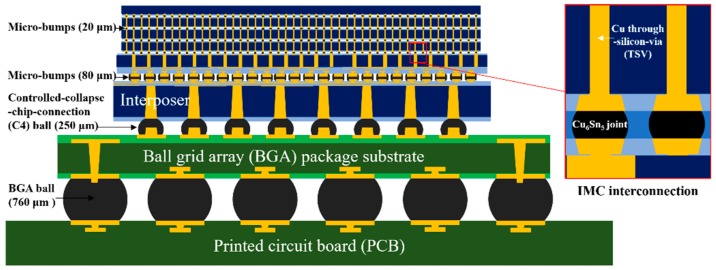
Schematic illustration of a 3D IC using 3D package technology. IMC = intermetallic phase.

**Figure 2 materials-12-04127-f002:**
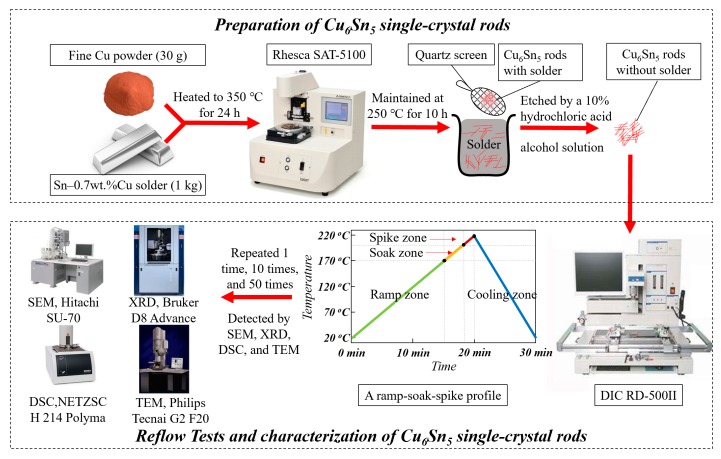
Flowchart of the preparation, reflow tests and the characterization of Cu_6_Sn_5_ IMCs. DSC = differential scanning calorimetry.

**Figure 3 materials-12-04127-f003:**
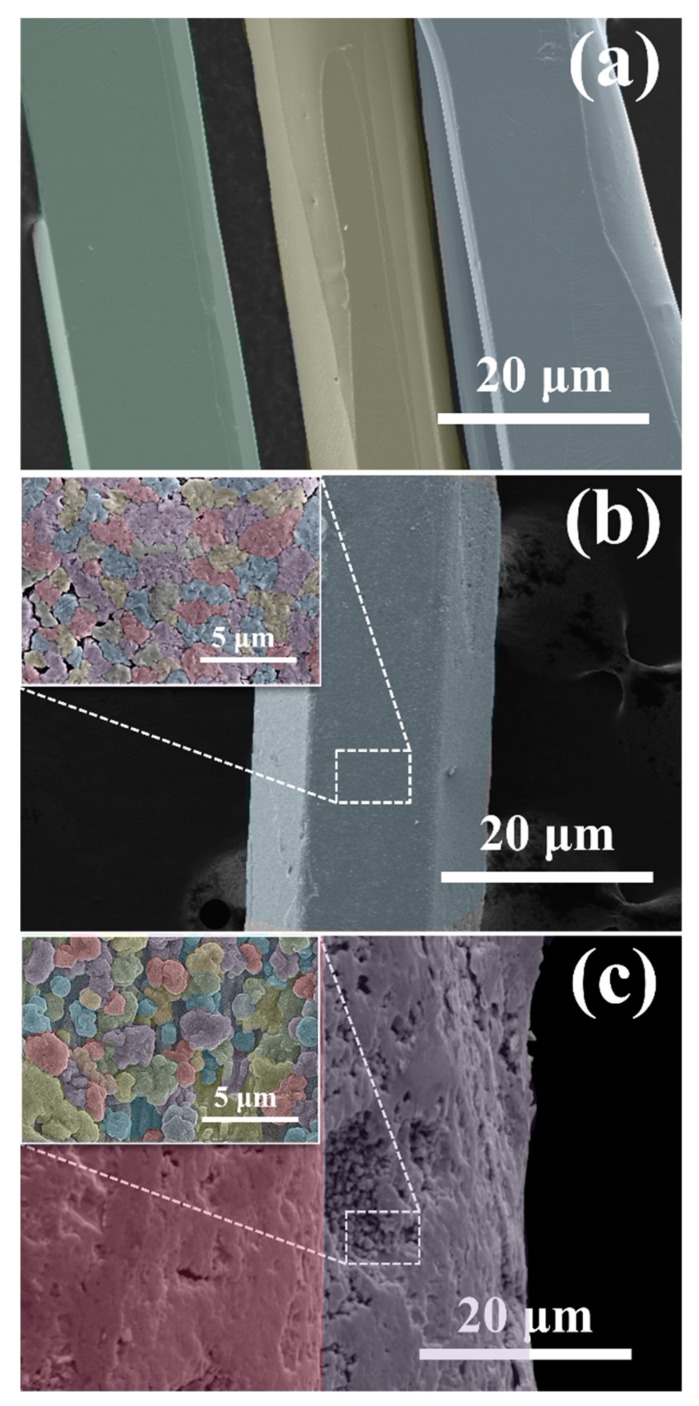
Surface microstructures of the Cu_6_Sn_5_ grains after being reflowed (**a**) 1 time; (**b**) 10 times; (**c**) 50 times.

**Figure 4 materials-12-04127-f004:**
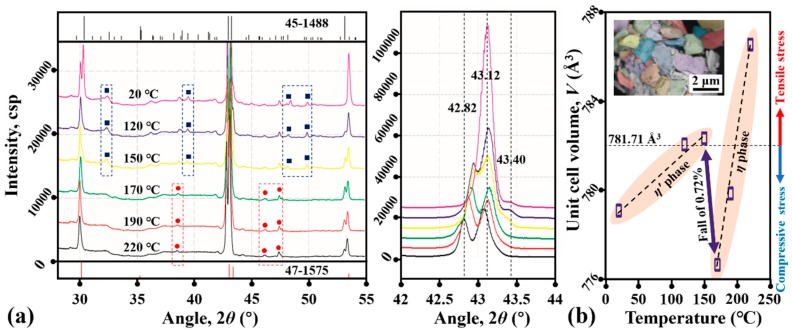
(**a**) In situ XRD patterns of the 1-time-reflow Cu_6_Sn_5_ powder in the 2*θ* range of 30°–55°; (**b**) the calculated unit cell volume (*V*) at selected temperatures. Note that the monoclinic structure of ICDD 45-1488 is used to fit the pattern in (**b**), and the microstructures of the Cu_6_Sn_5_ powder used in our XRD tests are also inserted in (**b**).

**Figure 5 materials-12-04127-f005:**
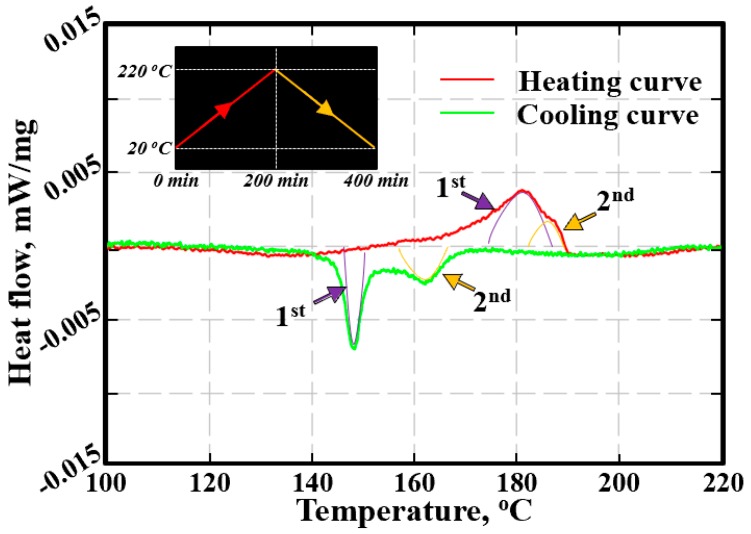
DSC heat flow curves of the Cu_6_Sn_5_ powder after being reflowed 1 time, and the pre-set temperature profile is also included.

**Figure 6 materials-12-04127-f006:**
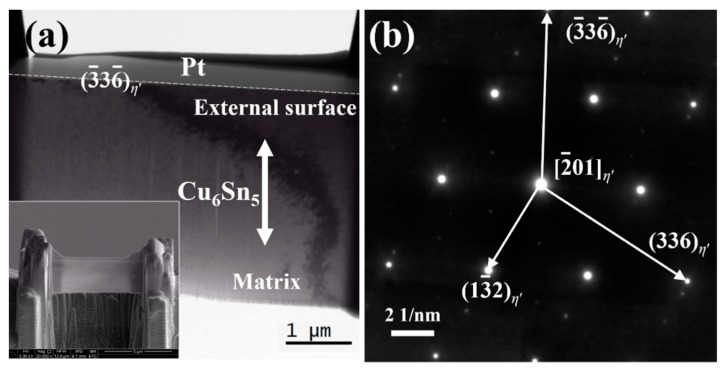
(**a**) TEM image of the cross-section microstructure for a rod-type Cu_6_Sn_5_ grain after being reflowed 1 time; (**b**) the corresponding surface area diffraction pattern (SADP) for the matrix. The SEM image of the sample after focused-ion beam (FIB) cutting is also included in (**a**).

**Figure 7 materials-12-04127-f007:**
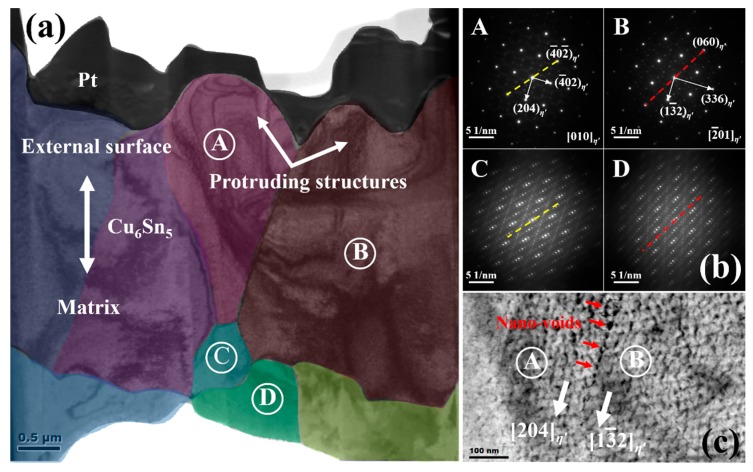
(**a**) TEM image of the cross-section microstructure of a rod-type Cu_6_Sn_5_ grain reflowed 50 times; (**b**) the SADPs for the four grains marked in (**a**); (**c**) an enlarged view of the local microstructure at the boundary between Grain A and Grain B.

**Figure 8 materials-12-04127-f008:**
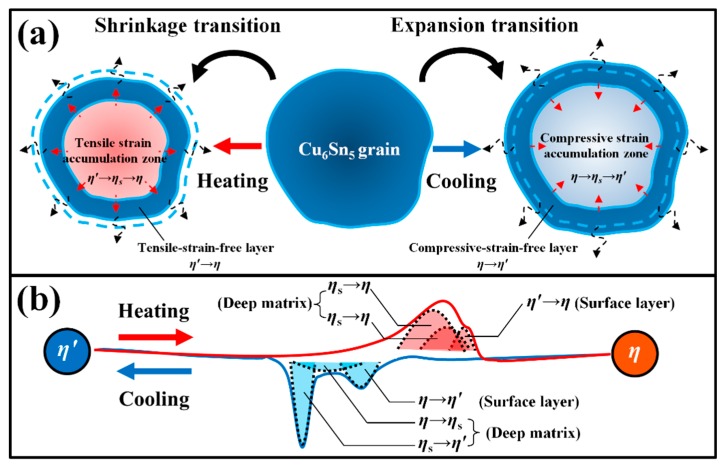
(**a**) Sketch map of the structure-induced mechanism of Cu_6_Sn_5_ polymorphic transition; (**b**) the reason for the distribution of latent heat peaks in Cu_6_Sn_5_ polymorphic transition.

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
