# Peer review of "Damage Mechanism of Cu6Sn5 Intermetallics Due to Cyclic Polymorphic Transitions"

_materials, 2019, doi:10.3390/ma12244127_

Round 1

Reviewer 1 Report

This paper presents results of investigations on a polymorphic transition in Cu6Sn5 grains after 1, 10 and 50 reflow cycles. For the reflow conditions, the peak temperature and cooling rate were set at 220 °C and 20 °C/min, respectively. The XRD measurement was also conducted at a cooling rate of 20 °C/min from 220 °C and the XRD results have indicated the transformation of η to η' occurred during cooling in the reflow process. The corresponding TEM analysis also confirms that the entire Cu6Sn5 grain has fully transformed into the η'-phase structure after 1 reflow cycle.

The Cu6Sn5 grains maintain a rod-like shape with flat and smooth surfaces after 1 reflow cycle. While, increasing the reflow cycles up to 50 times, they showed that the Cu6Sn5 surfaces become damaged and fragmented, and numerous protrusion-like structures with irregular shapes and sizes have emerged in certain depressed areas.

Based on the structural observations (SEM and TEM), it is reasonable to speculate that the multiple reflows would produce an adverse effect on the reliability of Cu6Sn5 interconnections in 3D package whenever the polymorphic phase transitions exist during the process. As for a future research recommendation, it will be interesting to see if this adverse effect can be controlled with different cooling rates in the reflow profile.  

They have also proposed the damage mechanism induced by the cyclic polymorphic transition in a single Cu6Sn5 grain based on the investigations results, which the reviewer agrees can provide in-depth understandings of the phase transformation behavior of Cu6Sn5.

The manuscript is well written, and the review suggests to accept the paper for publication.

Author Response

Thanks for your hard work, and early to wish you and your family a happy Thanksgiving Day!

Reviewer 2 Report

Dear Authors, I personally liked the paper very much - the manuscript is well written, the experiments are well thought, also the results are interesting and might add not only to scientific, but practical advancement of the topic as well. However there are a few comments worth considering in a minor revision.

- What is the practical approach of 10-50 reflow cycles? Please emphasize the theoretical approach on this cycle number, or emphasize more the connection to practical uses where such number of cycles might be important.
- Maybe use the word: lattice mismatch instead of misfit?
- Please do not "speculate", but maybe use "hypothesis" or other more scientific words to describe your proposed explanation.
- Please further emphasize practical applicability in the conclusions following the mention of 3D packages.

Author Response

Point 1:

- What is the practical approach of 10-50 reflow cycles? Please emphasize the theoretical approach on this cycle number, or emphasize more the connection to practical uses where such number of cycles might be important.

Response 1:

Thanks for the viewer’s professional suggestion.

In October 7th, 2019, Samsung Electronics, a world leader in advanced semiconductor technology, announced that it has developed an industry’s first 12-layer 3D-IC. Based on Samsung’s report, the stacking layers of chips in this 3D IC have been added from original 8 layers to 12 layers (Fig. R1). As we have mentioned in our manuscript, the traditional Sn-based micro-bumps (μ-Sn-bumps) in each chip layer must be transformed into the intermetallic (IMC) phase after the 1st reflow to avoid the collapse of μ-Sn-bumps in 3D packages. Therefore, to stack the 12 layers of chips, the μ-Sn-bumps on each chip layer must be completely transformed into the μ-IMC-joints after the 1st reflow. Accordingly, these μ-IMC-joints would experience repeated reflows during further chip stacking. For example, the μ-IMC-joints in the 1st layer of Samsung’s 12-layer 3D ICs would experience at least 11 reflow cycles. Because the number of stacking layers will never be an end, we predict that a 50-layer 3D IC may be developed a few years later, and the μ-IMC-joints in such 3D IC may experience over 50 reflow cycles. Hence, we conducted our experiments and study the reliability of Cu6Sn5 IMCs after 10 and 50 reflow cycles. We believe that our study can provide in-depth understandings of structural evolution and damage mechanism of Cu6Sn5 interconnections in real 3D package manufacturing.

Fig. R1 Cross-sectional structural changes between the 8-layer 3D IC and 12-layer 3D IC.

Following the reviewer’s advice, we add some descriptions to explain the reason of our study and experimental setup.

(Line 45)

In October 7th, 2019, Samsung Electronics announced that it has developed an industry’s first 12-layer 3D-Integrated-Circuit (3D IC), and the IMC joints in such 3D IC may experience over 10 reflow cycles; because the number of stacking layers will never be an end, a 50-layer 3D IC may be fabricated in the near future, and the IMC joints in such 3D IC may experience 50 (or even more) reflow cycles. Therefore, the determination of damage mode of Cu6Sn5 during multiple reflow processes is urgent in real 3D package manufacturing.

Point 2:

- Maybe use the word: lattice mismatch instead of misfit?

Response 2:

Thanks for the viewer’s professional suggestion. The word of “misfit” has been corrected by the word of “mismatch” in the revision.

Point 3:

- Please do not "speculate", but maybe use "hypothesis" or other more scientific words to describe your proposed explanation.

Response 3:

Thanks for the viewer’s professional suggestion. The word of “speculate” has been corrected by the word of “hypothesis” in the revision.

Point 4:

- Please further emphasize practical applicability in the conclusions following the mention of 3D packages.

Response 4:

Thanks for the viewer’s professional suggestion. Following your advice, we add some descriptions to emphasize practical applicability of our study in 3D package manufacturing.

(Line 215)

 It can be predicted that, if the Cu6Sn5 polymorphic transition cannot be limited effectively during multiple reflows, it may become one of the most dominate causes of failure for the Cu6Sn5 IMC joints in 3D packages.

Reviewer 3 Report

The paper surely deals with interesting problem relevant for Materials journal. The quality of some experiments, as well as of the applied equipment is very high. However, the overall presentation of the paper and treatment of the results is very poor. It is caused partially by the unclear terminology, but mainly by the experimental setup and the resulting conclusions also need to be justified. Please see my comments below:

The terminology is not clear. What do you mean by a "3D package“ and "interconnections“?

More details have to be provided about the preparation of Cu6Sn5 intermetallics and about the mentioned reflow process.

Have you checked whether the minor peaks in XRD patterns, which cannot be attributed to Cu6Sn5 phase do not belong to another Cu-Sn phase or any possible contamination?

How did you achieve the different colours of the grains in Fig. 1 and Fig. 5? Sice it is from the SEM and TEM, they have to be artificial.

In general, the presented data do not lead clearly to the stated conclusions (mentioned phase transformations, interpretation of the diffraction data etc.).

Author Response

Please read the uploaded file of 【the respond reviewer 3 -MDPI. DOCX】!

Reviewer 4 Report

Dear editor,

In the manuscript received, the authors study the structural evolution and damage mechanism of Cu6Sn5 interconnections. Despite the work is good and they seemed to be followed a correct research way, they have to improve drastically the way in which the paper has been performant. Thus, the manuscript should be accepted with major revision before being published in this Journal. In particular they have to improve:

The state of art. It is very small, they have to carry out a deep study of the literature and reflect it in the work with the aim of demonstrating that this study is a real advance on the current knowledge. The authors use informal words such as “scholars”, “our results”, “our previous work” …. Then, they have to use passive voice. Besides, the materials and methods section has to be improved too. They don’t show any graphic or picture of the work carried out…. The results and discussion from my point of view is ok. However, in the conclusions section they have to improve it in a way in which include topics that summarize and highlighting the results and contributions of the work.

Author Response

Please read the uploaded file of [Respond reviewer 4-MDPI. DOCX].

Reviewer 5 Report

As I can see from literature, there are n, n’, n6, n8 and n4+1 phases in Cu6Sn5.

http://dx.doi.org/10.1016/j.scriptamat.2012.12.012

Have you tried to compare your XRD patterns with such phases?

Page 3 Fig. 2.

Why the XRD pattern has been shown in the range of 30-55 degrees only? Why not from 14°?

As have been shown, for example, in http://dx.doi.org/10.1016/j.scriptamat.2012.12.012

Page 4

How do you think, why the peaks on DSC curves of your Cu6Sn5 are not agreement with, for example, published in

http://dx.doi.org/10.1557/JMR.2005.0371

Page 2 Lines 85-89

Moreover, there are numerous relatively weak peaks at 2θ values of 32.35°, 39.48°, 48.39° and 49.84° for the patterns in the temperature range of 20– 150 °C (marked by the dark-blue squares) and at 2θ values of 38.50°, 46.12° and 47.43° for the patterns in the temperature range of 170–220 °C (marked by the red circles), which cannot be well matched to the standard XRD files of the Cu6Sn5.

What about other CuxSny compounds? Have you tried to compare XRD profiles with, for example, Cu3Sn https://www.nature.com/articles/srep13491

Author Response

Please read the uploaded file of [Respond reviewer 5-MDPI. DOCX].

Reviewer 6 Report

Introduction. (1) At line 30, the explanation of “IMC” is not completely clear. Does “IMC” have the meaning of “intermetallic compound phase” or “intermetallic connection”? (2) For journal readers who are not expert in the field of the manuscript, the authors might consider adding a new figure that presents a schematic illustration of the 3D package, showing the Sn-based micro-bumps and the location of the intermetallic phase connections. This new figure would complement the present figures.

Materials and Methods. (1) At line 52, can information about the solder be provided in a parenthetical phrase? (2) At lines 53 – 55, can a reference be provided to support the experimental protocol presented in these two sentences? (3) In the second paragraph, some additional details might be provided: It would be helpful to include the approximate length of the eta Cu-Sn grains. (A diameter of 20 - 25 microns for these rod-shaped particles is stated at line 70 in the next section.) How many grains were ground into powder for the XRD and DSC analyses. Were special efforts needed to manipulate the small individual grains for FIB milling to prepare TEM specimens?

Results and Discussion. (1) At an appropriate place, the crystal structures of the two Cu-Sn phases (eta and eta-prime) should be noted. (2) Do the XRD patterns in Figure 2 (a) show evidence of preferred crystallographic orientation for the eta and eta-prime Cu-Sn phases? (3) At line 126, improved wording is needed for the phrase “to explore whether and how the existence of the”. (4) At lines 150 and 151, the lattice parameters are indicated to three places in Angstrom units. Does the experimental accuracy warrant this level of precision? A similar comment applies to “-4.71%” in line 153. Is the second decimal place warranted? (5) To place the experimental protocol in the practical manufacturing context, it should be noted now many reflows occur during the normal processing conditions. The study examined the scenario of 50 reflows. (6) Plausible comments have been provided about the phase transformation processes and their consequences, along with the occurrence of the putative intermediate eta-subscript “s” phase. What future experiments are recommended to show unambiguously that this phase does form? Limitations of the present study should be noted.

Author Response

Please read the uploaded file of [Respond reviwer 6- MDPI. DOCX].

Round 2

Reviewer 3 Report

All problems pointed out in my previous reviewer's report were solved or explained.

Author Response

(The authors gave the same response as above.)

Reviewer 4 Report

In general paper looks like good and sound enough. However the state of the art is some partial and continue being very small. The authors only added two references. The importance of the state of art has to be taken into account to test if the work carried out by them it is a real advance in the state of the art. Therefore, it is needed to be improved with the aim of when the manuscript will be published, the reader can confirm that the work is real novel research.Therefore, the manuscript has to be improved before the acceptance.

Regards.

Author Response

Please read the uploaded file of [Respond reviewer 4 - MDPI -round 2. docx]

Reviewer 5 Report

I'm satisfied with the author's answers.

Author Response

(The authors gave the same response as above.)
